# The deployment of a geomagnetic variometer station as auxiliary instrumentation for the study of Unidentified Aerial Phenomena

Foteini Vervelidou<sup>1</sup>, Alex Delacroix<sup>1</sup>, Laura Domine<sup>1,2</sup>, Ezra Kelderman<sup>1</sup>, Sarah Little<sup>1,3,4</sup>, Abraham Loeb<sup>1,2,†</sup>, Eric Masson<sup>1</sup>, Wesley Andres Watters<sup>1,4</sup>, and Abigail White<sup>1,2</sup>

**Correspondence:** Foteini Vervelidou (foteini.vervelidou@ext.esa.int)

Abstract. Witness reports of Unidentified Aerial Phenomena (UAP) occasionally associate UAP sightings with local electromagnetic interferences, such as spinning magnetic compasses onboard aircraft or sudden malfunctions of mechanical vehicles. These reports have motivated the incorporation of a magnetometer into the instrumentation suite of the Galileo Project (GP), a Harvard-led scientific collaboration whose aim is to collect and analyze multi-sensor data that collectively could help elucidate the nature of UAP. The goal of the GP magnetometry investigation is to identify magnetic anomalies that cannot be readily explained in terms of a natural or human-made origin, and analyze these jointly with the data collected from the other modalities. These include an ensemble of visible and infrared cameras, a broadband acoustic system and a weather-monitoring system. Here, we present GP's first geomagnetic variometer station, deployed at the GP observatory in Colorado, USA. We describe the calibration and deployment of the instrumentation, which consists of a vector magnetometer and its data acquisition system, and the collection and processing of the data. Moreover, we present and discuss examples of the magnetic field data obtained over a period of 6 months, including data recorded during the May 2024 G5 extreme geomagnetic storm. We find that the data meet and even surpass the requirements laid out in GP's Science Traceability Matrix. Key to the evaluation of our data is the proximity of the variometer station to the USGS magnetic observatory in Boulder, Colorado. By comparing the two sets of data, we find that they are of similar quality. Having established the proper functioning of the first GP variometer station, we will use it as the model for variometer stations at future GP observatories.

#### 1 Introduction

Modern-era sightings of Unidentified Aerial Phenomena (UAP) have been reported since the 1940s by both civilians and military personnel worldwide [e.g., Ruppelt (1956); von Reeken (1987); Amamiya (2009); Force (2010); Laurent et al. (2015)]. Despite the persistent and widespread nature of these sightings, instrumented and high-quality observations of these phenomena are very scarce, hindering investigation (Knuth et al., 2025). Recent initiatives such as NASA's commissioning of a UAP Independent Study team (NASA Unidentified Anomalous Phenomena Independent Study Team, 2024), the launch of the Sky

<sup>&</sup>lt;sup>1</sup> Galileo Project, 60 Garden Street, Cambridge, MA, USA 02138

<sup>&</sup>lt;sup>2</sup>Harvard-Smithsonian Center for Astrophysics, 60 Garden Street, Cambridge, MA, USA 02138

<sup>&</sup>lt;sup>3</sup>Scientific Coalition for UAP Studies, Fort Myers, FL 33913, USA

<sup>&</sup>lt;sup>4</sup>Whitin Observatory, Department of Physics & Astronomy, Wellesley College, 106 Central Street, Wellesley, MA 02481, USA

<sup>†</sup>Head of the Galileo Project

Canada Project (Sky Canada Project Team, 2025), and the call for an EU initiative on the study of UAP by 15 national organizations from 12 different countries (European Civil society UAP organisations, 2024), underline the increased public and governmental interest in working towards elucidating the origin and nature of UAP.

25

As with all scientific anomalies (i.e., observations that deviate from what is expected based on our current understanding of physical laws), the prerequisite for meaningful progress on the origin of UAP is the systematic and standardized collection of high-quality data. Given the enigmatic nature of UAP, a multi-sensor, multi-modal approach to data collection is the most appropriate, since it allows for the characterization of multiple physical properties of aerial objects and phenomena. For this reason, the Galileo Project has been assembling observatories equipped with optical, infrared, acoustic, weather-monitoring, and magnetic sensors (Watters et al., 2023).

The incorporation of a magnetometer in the Galileo Project multi-modal observatories is motivated by witness reports suggesting that UAP sightings are occasionally accompanied by strong magnetic field signals. Some of these reports describe isolated incidents, such as a 1953 UAP sighting in Yuma, Arizona (Maccabee, 2014). Here, the witness described seeing several uniformly-spaced concentric circles around a flying disk for as long as he had Polaroid glasses on, while the flying disk alone remained visible to him even with the glasses off. According to Maccabee (2014), a very strong magnetic field signal surrounding the disk could explain this incident, since a magnetic field aligned with the direction of light propagating through the Earth's atmosphere can cause the plane of polarization of linearly polarized light to rotate (Oberoi and Lonsdale, 2012; Meessen, 2012). This would lead to some light passing through the polarized glasses and some light getting blocked by them. Another isolated incident has been reported to have taken place in Florida in 1992 (Maccabee, 1994). According to this report, a day after an eyewitness spotted a UAP over the roof of her house, a close acquaintance of hers inspected the yard with a magnetic gradiometer and found a strong magnetic signal (18000-25000 nT/m), while no UAP was visually present anymore. The signal was purportedly pulsating at a rate of around 10 Hz. The next day the signal was weaker and three days later there was no detectable signal.

Beyond these singular reports, a recurring phenomenon is that of military and civilian pilots reporting perturbations in the aircraft's onboard magnetic compasses during encounters with UAP (Haines, 1992; Weinstein, 2012). Another recurring phenomenon are reports of electrical system failure in vehicles, such as headlight or car engine failures. It has been suggested that high frequency electromagnetic radiation, such as microwave radiation, could be the cause of these failures (Johnson, 1983, 1988; McCampbell, 1983; Powell et al., 2024).

A systematic approach to studying the magnetic effects of UAP has been followed by the Project Starlight International. In the framework of this project, uncalibrated magnetometers (in addition to radars, gravimeters and cameras) were installed in Texas and White Sands, New Mexico, in 1974 and 1978. According to Meessen (2012), in cases that UAP were seen and filmed, their magnetometers recorded spikes at 6 Hz in the spectrum of the magnetic signal. Moreover, spikes in intensity, at least 5 times that of the baseline measurements, were recorded when the visible objects reversed motion.

A more recent organized effort to document the magnetic signature of UAP concerns Project Hessdalen (Strand, 1984). This project was launched in 1983 and continues today with the aim of investigating the so-called "Hessdalen Lights", a recurring nocturnal light phenomenon of unknown origin. Over the years, multiple observational campaigns have been conducted, involv-

ing the deployment of a variety of instruments, including magnetometers. According to Teodorani (2004), some observations of these lights were accompanied by magnetic perturbations.

Electromagnetic perturbations in the presence of luminous objects of unknown origin have also been reported in a different location by Tedesco and Tedesco (2024). Over the course of ten months, Tedesco and Tedesco (2024) deployed on the south shore of Long Island, NY, a multi-modal instrumentation platform, which included electromagnetic field transducers. They reported peaks in the electromagnetic power flux density at frequencies of 1.8 GHz and 4 GHz during sightings of UAP.

Magnetic field anomalies occurring in the vicinity of UAP sightings have also been reported by Project Match, an ongoing collaboration between the Multiple Anomaly Detection and Automated Recording (MADAR) Project (https://madar.site) and the National UFO Reporting Center (NUFORC). MADAR, established in 1970, consists of a network of magnetometers deployed at various locations around the world, which trigger an alarm system when the readings exceed a certain, undocumented, threshold. According to NUFORC, which collects reports of UAP sightings, magnetic field anomalies recorded by MADAR have occurred in the vicinity of reported UAP sightings (NUFORC, 2023).

The aforementioned reports and findings, although resulting mainly from private citizen science efforts, have motivated us to conduct electromagnetic field measurements at the Galileo Project (GP) observatories. According to some of these reports [e.g., the ones related to car engine failures or the findings by Tedesco and Tedesco (2024)], these perturbations occur in the microwave region of the electromagnetic spectrum (1-300 GHz), while reports based on anomalous magnetometer readings and compasses deviations suggest that perturbations occur in the extremely low frequency range of the electromagnetic power spectrum (

**Figure 1.** The elements comprising the instrumentation system of our geomagnetic variometer station, assembled in our lab. The magnetometer is a three-axes fluxgate Bartington Instruments magnetometer and is powered by a Bartington Instruments power supply unit (PSU). The data acquisition module (DAQ) is a NI-9239 from National Instruments, which contains an analog-to-digital converter. The output digital data are collected by a NI cDAQ-9171 (chassis), which is connected via USB to a mini PC by Intel (not shown here). See text for more details.

The measurement range of the magnetometer is  $\pm$  100  $\mu$ T, its sensitivity is 1 V per 10  $\mu$ T, its measurement noise floor is  $\leq$  10 pTrms/ $\sqrt{\rm Hz}$  at 1 Hz, and its frequency response spans from DC to 1 kHz (-3dB at 3 kHz). It is connected to the PSU by a 10-meter cable, also supplied by Bartington Instruments. The PSU has a noise floor of < 5pT/ $\sqrt{\rm Hz}$  at 1 Hz. The output of the PSU is three BNC cables, one for each magnetic field component. These BNC cables connect to the DAQ. The DAQ contains a 24-bit delta-sigma analog-to-digital converter and has an input range of  $\pm$  10 V. The DAQ allows for sampling rates,  $f_s$ , that range from 1.613 to 50 kHz, has an alias-free bandwidth of  $0.453 \times f_s$ , and an input-referred noise of 70  $\mu V_{rms}$  or equivalently 700 pT. Assuming that the input-referred noise corresponds to the highest possible sampling rate, the expected noise in our raw measurements, given that we sample at the lowest possible  $f_s$ , is  $\approx$ 120 pT. This makes the noise floor of the DAQ the main contributor to the expected noise of our setup. The DAQ is mounted on the chassis and the chassis is connected via USB to the NUC. Figure 1 shows the various elements of the instrumentation setup connected to each other, except for the NUC.

The magnetometer has an integrated temperature sensor but the magnetometer cable transfers only the magnetic field readings to the PSU and not the temperature readings. To access the temperature data, we opened the magnetometer cable and connected its temperature pin to the DAQ using a custom-made cable. Moreover, we tied the magnetometer and DAQ grounds together to ensure a common electrical reference point between the two devices. With this cabling, we obtained accurate temperature readings during the first three months after deployment (March 28-June 30, 2024). On July 1st, 2024, the temperature readings became erratic. Meanwhile, we were informed by Bartington Instruments that the integrated temperature sensor fea-

ture has been removed from future Mag-13MS100 magnetometers. Planning ahead, we decided to not interrupt the recordings to troubleshoot the issue and to rather install an independent temperature and humidity sensor external to the magnetometer, the same that we would be using at future sites. Unfortunately, the new sensor failed to collect measurements (see the Appendix for details). Since we did not resolve the issue with the integrated temperature sensor either, we only collected temperature measurements until June 30, 2024.

#### 3 Data recording and storage

Our first tests of the magnetometer setup (results not shown here) were conducted by using the National Instruments (NI) Graphic User Interface (GUI) software NI-DAQ<sup>TM</sup>mx (National Instruments, 2023). This software allows for real-time monitoring of the recordings and stores the data in .cvs or .tdms files. This software was used also to perform the temperature calibration of the magnetometer (see section 4) and to orient the magnetometer after its deployment at our site (see section 5). To obtain long-term, continuous recordings, we developed our own script, using Python (Domine and White, 2025). We performed the recordings at the lowest sampling rate permitted by our DAQ, which is 1612.9 Hz. It is worth mentioning that this DAQ rejects out-of-band signals for the selected sampling rate by means of a combination of analog and digital filtering. Given our sampling frequency of 1612.9 Hz, this means that our raw data contain frequencies up to 806 Hz, which lie within the frequency range of our magnetometer.

We stored the data into one-hour files. The data files were saved temporarily in the NUC, whose solid-state drive (SSD) has a 1 TB storage space. By means of an external hard disk, the data were periodically transferred to Harvard's computing cluster for long-term storage and processing. Moreover, as a back-up, the data have also been stored in a Network Attached Storage (NAS) device.

# 4 Calibration

160

Readings of vector magnetometers depend strongly on the ambient temperature. Our magnetometer has been delivered to us tested by Bartington Instruments at T=19.9 °C and T=22.3 °C. This means that within this temperature range the magnetometer provides absolute magnetic field measurements within specifications, as long as its baseline drift remains negligible. However, whenever the magnetometer is exposed to temperatures outside this temperature range, for the measurements to be absolute, they would have to be corrected for the effect of the temperature variation. This requires establishing the relationship between magnetic field readings and temperature. This relationship is unique for each magnetometer and can be determined through a calibration process, in which a scalar magnetometer acts as the point of reference. We calibrated our magnetometer at the magnetic observatory of the United States Geological Survey (USGS) in Boulder, Colorado, in collaboration with its personnel. This observatory is part of the international INTERMAGNET network, which means that it delivers data of the highest quality, in line with the stringent criteria established by the geomagnetic scientific community (Love and Chulliat, 2013).

To perform the calibration, we followed the protocol by Merayo et al. (2000). According to this protocol, the calibrated measurements along the X (North), Y (East), and Z (Down) axes,  $X_{cal}$ ,  $Y_{cal}$ , and  $Z_{cal}$ , respectively, are obtained by the raw measurements through the following expression:

$$\begin{bmatrix} X_{cal} \\ Y_{cal} \\ Z_{cal} \end{bmatrix} = A \times \begin{bmatrix} X_{raw} - O_1 \\ Y_{raw} - O_2 \\ Z_{raw} - O_3 \end{bmatrix}, \tag{1}$$

where  $X_{raw}$ ,  $Y_{raw}$ , and  $Z_{raw}$  are the raw measurements along the X, Y, and Z axes, respectively, the matrix A is given by

$$A = \begin{bmatrix} a_{11} & a_{12} & a_{13} \\ 0 & a_{22} & a_{23} \\ 0 & 0 & a_{33} \end{bmatrix}, \tag{2}$$

with  $a_{11}$ ,  $a_{22}$ ,  $a_{33}$  the scaling coefficients, and  $a_{12}$ ,  $a_{13}$ ,  $a_{23}$  the orthogonality coefficients, and the vector O is given by

$$O = \begin{bmatrix} O_1 \\ O_2 \\ O_3 \end{bmatrix}$$
, (3)

with  $O_1$ ,  $O_2$ ,  $O_3$  the offset coefficients. To obtain the final scaling and orthogonality coefficients, the elements of the matrix A need to be rescaled by the value recorded by the scalar magnetometer that was used as a reference point.

The aim of the calibration is to determine the scaling, orthogonality and offset coefficients as a function of temperature. Figure 2 shows our setup for the calibration process. Our magnetometer was attached to a theodolite by means of a 3D printed mount, as shown in Figure 2a. Our magnetometer mounted on the theodolite and a scalar magnetometer mounted on a tripod were placed inside a climate-controlled chamber, as shown in Figure 2b. After setting the climate-controlled chamber at a given temperature, we rotated our magnetometer with the aid of the theodolite, while recording the measurements. We rotated it at 5° increments around the horizontal axis and at each position we made a full turn around the vertical axis.

According to our initial calibration plan, we would collect measurements at temperatures spanning the range between 5 °C and 35 °C, the maximum temperature range we expected our magnetometer to be exposed to, once buried underground. Unfortunately, due to a malfunction of the AC system, it was only possible to heat the climate-controlled chamber but not to cool it down. Therefore, we collected measurements at 23 °C, 28 °C and 33 °C. The vector measurements collected at a given temperature during a full 3D rotation of the magnetometer represented the  $X_{raw}$ ,  $Y_{raw}$ , and  $Z_{raw}$  of Equation 1. The scaling, orthogonality and offset coefficients of Equation 1 were obtained by means of an inversion script written by Alain Barraud and Suzanne Lesecq (Barraud and Lesecq, 2008). The mean value of the readings of the scalar magnetometer during the acquisition of the vector measurements, corrected for the constant offset between the scalar and vector magnetometers due to the  $\approx$  3 meters distance between them, was used to rescale the scaling and orthogonality coefficients to their final values.

**Figure 2.** The measurement setup during the temperature calibration of our magnetometer at the USGS Boulder magnetic observatory. (a) Our vector magnetometer mounted on a theodolite by means of a 3D printed mount. (b) The vector magnetometer mounted on the theodolite, and the scalar magnetometer mounted on a tripod, inside the climate-controlled chamber.

**Figure 3.** The calibration coefficients of our vector magnetometer, as a function of temperature. (a-c) The offset coefficients. (d-f) The scaling coefficients. (h-i) The orthogonality coefficients. Blue circles correspond to the values we obtained for 23  $^{\circ}$ C, 28  $^{\circ}$ C and 33  $^{\circ}$ C. The red solid lines correspond to the least squares best-fit lines. The dashed black lines correspond to  $1-\sigma$  uncertainties derived from the least squares fit.

Figure 3 shows our results (blue circles) along with the best-fit lines (red solid lines) derived via least-squares. These best-fit lines can be used to estimate the values of each of these coefficients at any temperature within this temperature range. Shown are also the corresponding  $1-\sigma$  uncertainties from the least squares fit, calculated as the square root of the diagonal elements of the covariance matrix (black dashed lines). These uncertainties translate into tens of nT uncertainty for the calibrated magnetic field measurements. However, these are highly conservative estimates, as demonstrated by Figure 4, which shows the results of an empirical evaluation of our calibration.

For this evaluation, we run simultaneously our vector magnetometer and the scalar magnetometer, while allowing the ambient temperature to vary naturally. The results we obtained by measuring over the span of one hour (duration imposed by logistical constraints) are shown in Figure 4. The intensity of the uncalibrated vector data are shown in black, and the intensity of the calibrated data are shown in magenta. The scalar data are shown in orange, the scalar data with their baseline adjusted to that of the intensity of the uncalibrated vector data are shown in light blue, and the scalar data with their baseline adjusted to that of the intensity of the calibrated vector data are shown in blue. The maximum and minimum values of the intensity of the calibrated vector data, accounting for the 1- $\sigma$  least-squares uncertainties of the calibration coefficients shown in Figure 3, are shown in grey. While these values allow for a  $\pm$  15 nT uncertainty, we see that the baseline of the calibrated data matches closely that of the scalar data (the difference decreased from  $\approx$  40 nT to  $\approx$  2 nT) and, more importantly, the slope of the calibrated data matches almost perfectly that of the scalar data (maximum difference decreased from 5 nT to <1 nT).

## 5 Deployment

For the deployment of our variometer, we selected a site without magnetic interference from human-made sources. We also took steps to protect all the instrumentation from exposure to water and intense sunlight. Moreover, the magnetometer itself was protected from being exposed to strong temperature variations and its proper orientation was ensured by means of a custom-made mount. Importantly, none of the items used for the deployment (e.g., screws) were magnetic. In the following, we provide details about how we implemented the above during the deployment of our variometer station at a private site in Boulder, Colorado.

The site lies within a 160,000 m<sup>2</sup> horse ranch. The horses did not graze within 75 meters of our site, for the entire duration of the data collection period (March 28-September 26, 2024). A highway runs NE/SW at a distance of approximately 160 meters from the site, and this is also where the nearest building (i.e., a barn with adjacent hay storage) is located. The power was provided from solar panels on top of the lab space, stored in batteries within the lab space. Internet access was established through Starlink and 5G cellular equipment installed at the site. The magnetometer was installed at the south corner of the site, while the data acquisition system (i.e., the PSU, the DAQ, the chassis and the NUC) were installed 5 meters away to the west (see Figure 5). The items of the data acquisition system were mounted on a custom 3D-printed plastic support plate and were secured with Velcro straps through slots in the plate or bolted to potted inserts. Special attention was paid to the proper organization of the cables, such that they would not give rise to interference. The support plate with the electronics was

Figure 4. The results of an empirical evaluation of our temperature calibration, during a 1-hour recording under natural temperature variation. (a) The intensity of the raw uncalibrated data, plotted over time. (b) The results of the calibration. Black line: the intensity of the 1-sec uncalibrated vector data. Magenta line: the intensity of the calibrated vector data. Grey lines: the maximum and minimum values of the intensity of the calibrated vector data, accounting for the 1- $\sigma$  least-squares uncertainties of the calibration coefficients shown in Figure 3. Orange line: the scalar data. Light blue line: the scalar data, with their baseline adjusted to that of the intensity of the uncalibrated vector data. Blue line: the scalar data, with the baseline adjusted to that of the calibrated vector data.

**Figure 5.** Diagram of the deployment site, showing the location of various instruments, including the magnetometer, and the distance among them. The blue circle shows the location of the hole, inside which the magnetometer was deployed. The blue rectangular shows the location of the magnetometer's data acquisition system. The yellow triangle and cyan square show the location of the Pan-Tilt-Zoom Beacon 8.0 camera (PTZ) and the all-sky camera arrays (MD) (Domine et al., 2025), respectively, which were also deployed at the site. The light yellow rectangular shows the location of the lab space.

housed in a waterproof, plastic enclosure, installed inside a fiberglass shade to protect against sunlight. The entire assembly was mounted on wooden posts (see Figure 6).

To minimize exposure to temperature variations, the magnetometer sensor was placed underground. For this, we dug a 1 meter deep hole, covered the bottom with concrete, and placed on it a 1 meter long plastic inspection chamber. The magnetometer sensor was placed inside a plastic bucket and the bucket was placed inside the inspection chamber. This configuration made the magnetometer accessible to us, even after we closed the hole with dirt. To ensure the proper orientation of the magnetometer sensor, the sensor was installed on a custom-made mount, shown in Figure 7. This mount was designed to allow for tip and tilt adjustment and rotation around the sensor's vertical axis. Additionally, the mount includes a bubble level. This mount was bonded to the bottom of the bucket by means of double-sided adhesive. All the materials used for the mount were non-magnetic. The bucket and the inspection chamber each had a hole on the side, where waterproof bulkheads were fitted securely. These held a plastic conduit, through which the magnetometer cable was routed into a 5 meters long trench that led to the electronics enclosure (see Figure 8). The bucket was sealed with a waterproof lid. Both the lid and the interior wall of the bucket were lined with thermal insulation. For additional insulation, one blanket was wrapped around the plastic conduit exiting the inspection chamber and another one filled the void inside the inspection chamber. Furthermore, eighteen 1 liter

**Figure 6.** The waterproof, plastic enclosure with the data acquisition system of the magnetometer, mounted on wooden posts and covered by a sunshade.

water bottles were placed around the perimeter of the chamber to further minimize temperature variations, and were packed in with dirt to keep them in position. Any remaining space around the chamber was also refilled with dirt.

To properly orient the magnetometer we proceeded as follows. The custom-made mount was bonded to the bottom of the bucket, aligned horizontally by means of the bubble level and the tip-tilt adjusters. Then the magnetometer was mounted on it so that its Z axis was pointing downwards, perpendicular to the ground. Subsequently, the magnetometer was rotated around its Z axis until the readings along its Y axis became zero. This meant that the X axis of the magnetometer was pointing toward magnetic North. Finally, we corrected for the magnetic declination so that the X axis was pointing toward geographic North. The magnetic declination of the site (i.e., the angle between magnetic North and geographic North) was obtained by means of the online NOAA Magnetic Declination Calculator, based on the US/UK World Magnetic Model (WMM) for 2020-2025 (Chulliat et al., 2020).

To ensure that the horses of the ranch would not approach our instruments, we deployed an electric fence around our site (yellow line in Figure 5). As we discuss in the next section, a number of tests ensured that this fence was not affecting our measurements.

**Figure 7.** The custom-made mount for the magnetometer. (a) A schematic diagram showing the mount inside the bucket. (b) A picture of the magnetometer placed on the mount.

## 6 Results

On March 28, 2024, we oriented the magnetometer, turned the fence on, and started recording. The magnetometer remained deployed until September 26, 2024. During this 6 month period, we recorded data except at times when the system ran out of power and until it was switched back on. In this section, we present examples of both raw data (sampling rate of 1612.9 Hz) and 1-second values. The latter serve as a means of comparison between our data and the data provided by the nearest USGS magnetic observatory, located in Boulder, Colorado, at about 60 km distance from our site. Its 1-minute and 1-second data are open access and are available at the INTERMAGNET website (https://intermagnet.org; the observatory's codename is BOU). For this comparison, we applied a 1-second moving average window on our raw data and we down-sampled by selecting 1 out of 1613 data points. Moreover, we adjusted our baseline to that of BOU, since we are only interested in evaluating our ability to accurately record magnetic field variations.

We present data obtained during magnetically quiet and magnetically disturbed days. For this, we rely on the Kp index, which is used to quantify disturbances of Earth's magnetic field caused by solar storms. Values of Kp vary between 0 and 9, with  $Kp \ge 5$  signaling disturbed conditions, known as magnetic storms, and Kp=9 signaling an extreme magnetic storm.

**Figure 8.** The magnetometer buried underground. (a) A picture of the hole in which the magnetometer was deployed. Shown are the yellow inspection chamber, the bucket with its lid open, the magnetometer on the mount, and the trench enclosing the magnetometer cable inside a plastic conduit. The water bottles, placed vertically, surround the hole and are covered in dirt, therefore they are not visible. (b) The hole and the trench filled with dirt, the inspection chamber with its lid closed and the electronics enclosure.

The Kp index is calculated and made publicly available (https://kp.gfz.de/en/) by the GFZ Helmholtz Centre for Geosciences (Matzka et al., 2021).

#### 6.1 Magnetically quiet days

Figure 9 shows the raw measurements (left column) and the corresponding 1-sec data (right column) obtained during May 9th, 2024, a magnetically quiet day (Kp index 

**Figure 9.** Vector magnetic field data recorded on May 9, 2024, a magnetically quiet day (Kp

**Figure 10.** Raw temperature data, recorded with the integrated temperature sensor on May 9, 2024, with a sampling rate of 1612.9 Hz. The inset shows a 300 data points zoom-in. Temperature readings vary by less than 0.3 °C, as a result of the thermal insulation measures we took during deployment. Similar diurnal temperature variations characterize all our data up until July 1st, which is when the temperature readings became erratic (see text for details).

according to the specifications of the magnetometer and the DAQ, presented in section 2. The potential source of this noise is discussed in the Discussion section. Given the experimental noise of our raw measurements, the expected RMS noise of the 1-sec data is  $\approx 1\text{-}2$  nT per magnetic field component, which is indeed what we observe. In addition to that, we observe that the Z component of our raw measurements (Figure 9e) exhibits a  $\approx 40$  s long spike with an amplitude of  $\approx 80$  nT. As will be discussed in section 6.4 and in the Discussion section, such spikes occur in our raw measurements usually a few times per week, at irregular intervals.

Figure 10 shows the raw temperature measurements, obtained with a sampling rate of 1612.9 Hz by the integrated temperature sensor of our magnetometer, during the day of May 9th, 2024. The inset shows a 300 data points zoom-in. We see that the temperature throughout the day varies by less than  $0.3\,^{\circ}$ C. All temperature recordings, up until July 1st when the temperature recordings became erratic (>300 $^{\circ}$ C), were characterized by the same diurnal stability. This is the result of the thermal insulation measures we took during deployment. Given the lack of temperature variations during a given day, we did not apply any temperature calibration to our magnetic field measurements. We note that the RMS noise in the raw temperature data shown in Figure 10 is  $\approx 0.07\,^{\circ}$ C. Although we do not have information about the noise specifications of the integrated temperature sensor, the observed noise level is consistent with typical industrial temperature sensors.

**Figure 11.** Magnetic field data during the geomagnetic storm on May 10-12, 2024. (a) The magnetic field component pointing Down (Z), from noon on May 10 to 9 am on May 12, in UTC time. Black solid line shows our 1-sec magnetic field data, after adjusting the baseline to BOU's baseline. Red dashed line shows the 1-sec magnetic field data recorded at BOU. Note that our magnetometer ran out of power at 9 pm on May 10 and started recording again on 9 pm on May 11. (b) The Kp index values for the same time interval as shown in (a). Green: Kp

**Figure 12.** Magnetic field data over two days with controlled human-made electromagnetic interference. Our 1-sec magnetic field data, after their baseline has been adjusted to that of BOU, are shown in black solid line. The 1-sec magnetic field data recorded at BOU are shown in red dashed line. (a) The magnetic field component pointing Down (Z), on June 15, 2024. Note the two spikes in our data at 17:40 and 17:48 UTC (marked by arrows), due to the spraying of the electric fence with water. (b) The magnetic field component pointing Down (Z), on September 18, 2024. Note the spike in our data at 19:59 UTC, due to the use of a steel tool next to the magnetometer, and the subsequent smaller spikes and loss of the baseline (see text for details).

Figure 13. Installation of ferrites around the cables to suppress high frequency electronic noise.

appeared at irregular intervals, usually a few times per week. Although we did not seek to eliminate them in post-processing, the 1-second moving average and down-sampling removed most of the spikes and the remaining ones were attenuated. We tried to eliminate these spikes by installing ferrites (i.e., magnetic components that suppress high-frequency noise) around the cables inside the electronics enclosure. Since the ferrites recommended by NI to be used in tandem with our DAQ were not readily available, we used generic ones. On September 7th, we installed five ferrites, one around each of the four BNC cables plugging into the DAQ, and one around the magnetometer cable, as shown in Fig. 13. Nevertheless, the spikes kept occurring. Fig. 14 shows the data we acquired on September 9, 2024. Figs 14a and b show our raw and 1-sec data, respectively, along the X component. The raw data have a  $\approx 20$  nT spike, of 40 ms duration, at about 13:45 UTC. This spike is not visible in the 1-sec data. Figs 14c and d show our raw and 1-sec data, respectively, along the Z component. The raw data have a  $\approx 500$  nT spike at the same time as component X. This spike leaked in the 1-sec data, in the form of two spikes, occurring about 10 minutes before and 10 minutes after the spike in the raw data. The earlier spike has an amplitude of  $\approx 2$  nT, while the amplitude of the later one is less than 1 nT.

# 7 Discussion

As noted in section 6.1, the experimental noise in our magnetic field recordings is two orders of magnitude larger than the theoretical noise floor, according to the specs of the magnetometer, the PSU1 and the DAQ. This theoretical noise floor, however, does not account for the 10 meters cable that we use to connect our magnetometer to the PSU1. Given our experience with the external temperature and humidity sensor (see Appendix A), we consider this to be the most plausible explanation for this discrepancy. Nevertheless, this experimental noise, which for our 1-sec data corresponds to 1-2 nT per magnetic

Figure 14. Our magnetic field measurements showing noise in the form of spikes (see text for details). (a) Raw (i.e., unprocessed) measurements of the magnetic field component pointing North (X) on September 9, 2024. Note the spike at 13:56:39 UTC (marked by the arrow). The inset shows a 1 second long extract centered on the spike. (b) The magnetic field component pointing North (X) of 1-sec data. Black solid line shows our magnetic field data, after their baseline has been adjusted to that of BOU. The moving average filtering eliminated the spike. Red dashed line shows 1-sec magnetic field data recorded at BOU. (c-d) Same as (a-b) for the magnetic field component pointing Down (Z). Note that the moving average filtering did not eliminate the spike but only reduced its amplitude.

field component, is still largely within our performance requirements. These were laid out in the GP Science Traceability Matrix (Watters et al., 2023), which states that the magnetometer should have a "resolution of order ~nT to resolve diurnal geomagnetic field variations". The diurnal variation, which occurs in magnetically quiet days mainly due to the ionospheric wind dynamo (Richmond et al., 1976), has typically an amplitude of several tens of nT. An example of this feature is clearly seen in Figure 9. Interestingly, our data not only capture this variation pattern in agreement with the data by BOU, but they even match the BOU data at the level of the few nT variations occurring along the Z component at 18:15:44 UTC. The fact that we can readily detect in our June 15 data (see Figure 12a) the 3 nT effect of spraying with water the electric fence surrounding the site further demonstrates that our system can resolve anomalies on the order of few nT. This exceeds the magnetometer's performance requirements by one order of magnitude. Overall, as showcased in Figures 9, 11, 12 and 14, our data captures the same features as that of BOU. This indicates that, at least under the conditions we encountered at this site, the quality of our data is similar to that of an INTERMAGNET observatory.

Our magnetometer also accurately records the large magnetic field variations that occur during geomagnetic storms. In Figure 11, we see that the May 2024 geomagnetic storm gave rise to several hundreds of nT magnetic field variations in the Z component, one order of magnitude larger than during magnetically quiet days. Unfortunately, we were able to record only the initial and last phases of the storm, which were magnetically quieter, because our station ran out of power in between. Nevertheless, we still captured the 100 nT increase in the magnetic field strength of the Z component between 19:00 and 21:00 UTC on May 10.

The ability of our magnetometer to accurately record large magnetic anomalies is also demonstrated in the data of September 18, when we made use of a 15 kg steel fence post driver next to the magnetometer. As mentioned above, this resulted in spikes of several hundred nT in all three magnetic field components, in both the raw and 1-sec data (see Figure 12b for the Z component of the 1-sec data), during the two minutes that the tool was used right next to where the magnetometer was buried. The subsequent smaller spikes are probably the magnetic signature of the fence post driver being used around the perimeter of the fence, while still in the vicinity of the magnetometer. Concerning the observed shift in our baseline, while we are uncertain about what caused it, our hypothesis is that the use of the fence post driver made the ground vibrate, which might resulted in a slight displacement of the magnetometer.

The installation of an electric fence around our instruments was clearly not ideal. This was necessary, however, to keep the horses of the ranch away from our instruments. Initially, the posts of the fence were plastic. However, after elks ran over and dismantled part of the fence, the plastic posts were replaced by steel posts, to reinforce the fence's stability. This replacement took place on April 30, 2024, and resulted in a steel post being located 2 meters away from the magnetometer hole. Despite this change, the quality of our data did not deteriorate and remained in good agreement with the BOU data, as showcased in Figures 9-12 and 14, which show data recorded after April 30. Through a series of tests, it was also established that the fence was not the source of the spikes occasionally appearing in our raw data, whose amplitude reaches up to hundreds of nT. In particular, on May 6, May 24, July 12, and September 18 we power cycled the fence and/or activated it by touching it, and none of these actions resulted in spikes in our data. On June 15, we sprayed the fence with water to test whether rainfall on the fence could give rise to spikes by causing leakage currents. The fact that at the times of the spraying we observed only two ≈3 nT spikes

in the Z component of the 1-sec data and no spikes in the raw data of any component indicates that humidity and rainfall could not account for the large spikes in our raw data. A final proof was that the spikes persisted even after the fence was permanently shut down on September 18, at 20:37 UTC. Having established that the electric fence was not the source of these spikes, we attempted to tackle the issue by installing ferrites around the cables in the electronics enclosure. Unfortunately, the spikes were still not eliminated.

There are many different sources that can give rise to spikes in magnetometer data, and it is possible that not all of the spikes in our data have a common source. The spikes that have a duration of less than a second are most probably not of natural origin, but longer spikes can also be of artificial origin, like digital errors. For example, we see in Figure 9e that the spike occurs just after a change in the gain. Some spikes could be due to electromagnetic interferences, which themselves have a large variety of causes. The road at 160 m distance from our site could be the source of some interference, especially whenever large vehicles like trucks were passing by. While we are not able to interpret all spikes in our raw data, we expect that some will be eliminated after we install the ferrites recommended by NI for electromagnetic compatibility compliance when using NI-9239 (i.e., our DAQ), as opposed to the generic, cost-effective ferrites we used in this deployment.

Future sites might require additional adjustments, both in the way we deploy our magnetometer and the way we process our data. This site had the advantage of being in a magnetically quiet environment, with no human-made magnetic field sources in the vicinity. Moreover, it had the additional advantage of lying close (at  $\approx 60$  km distance) to an INTERMAGNET observatory (Love and Chulliat, 2013). These characteristics made it an ideal site for testing our magnetometer and its deployment protocol. For the comparison between our data and the data by BOU, we relied on 1-sec values, given that BOU data of higher frequency are not publicly available. Going forward, we will continue to record and analyze our magnetic field measurements within their entire bandwidth of 0 to 806 Hz, given the  $\approx 10$  Hz magnetic signals reported during UAP sightings (Maccabee, 1994; Meessen, 2012) and the general need for more magnetic field data in UAP-related research efforts. The magnetic field amplitude of interest is harder to constrain based on past reports. Our magnetometer's 100 µT measurement range allows us to record anomalies as high as  $\approx 50 \,\mu\text{T}$  (the exact value depends on Earth's local magnetic field intensity), and the experimental noise level suggests that we can detect anomalies as low as few nT at 1 Hz in environments as magnetically quiet as our site in Colorado. Given that most UAP reports describe objects passing the observer at various distances, and magnetic field strength decreases for increasing distance from the magnetic field source, being able to record data spanning four orders of magnitude aims at enhancing our chances of recording signals of interest. The identification of potential anomalies in our magnetic field data will rely on the collection of magnetic field measurements over a sufficiently long time for us to gain an understanding of the typical magnetic field variations at each site, due to natural phenomena and human activity. Moreover, our magnetic field recordings will be compared with the magnetic field data of the closest available magnetic field observatories, and will be analyzed in tandem with the data collected by the multi-sensor, multi-modal instrumentation suite at the respective GP site.

#### 405 8 Conclusions

430

We presented the first geomagnetic variometer station deployed within the context of the Galileo Project for the study of UAP. Our instrumentation consists of a three-axes vector fluxgate magnetometer with an integrated temperature sensor, and a data acquisition system. We recorded magnetic field and temperature data using a Python script, and we stored the data in multiple hard disks, a Network Attached Storage (NAS) device, and Harvard's computing cluster. Our magnetometer was calibrated for temperature variations at the facilities of the USGS magnetic observatory in Boulder (BOU), in collaboration with members of its personnel. The deployment of our station took place in a magnetically quiet private site, 60 km away from BOU. During the six months of data collection, we recorded data during magnetically quiet and magnetically disturbed days, as well as in the presence of controlled magnetic interference. Our data captured clearly the diurnal variation of Earth's magnetic field during magnetically quiet days and recorded higher intensities during magnetic storms and controlled magnetic interference.

Beyond these indications that our magnetometer performs within requirements, we were able to evaluate the quality of our data by direct comparison to the data collected by BOU, an INTERMAGNET observatory. This comparison revealed that the collected data are of similar quality to those of BOU, which not only meets but surpasses the requirements for our project, given our purpose of detecting strong magnetic anomalies. This variometer station will serve as the blueprint for future variometer stations deployed at GP observatories.

Code and data availability. The python script used to record the magnetic field and temperature data is available at Zenodo (Domine and White, 2025). The raw magnetic field and temperature data discussed in Figures 3, 4, 9-12 and 14 are available at Zenodo (The Galileo Project, 2025). The data of the BOU magnetic observatory used in this study are available at the website of INTERMAGNET: https://intermagnet.org.

#### Appendix A: External temperature and humidity sensor

The integrated temperature sensor of the Mag-13MS100 magnetometer started giving erratic readings on July 1st, 2024. Moreover, we were informed by Bartington Instruments that upcoming versions of Mag-13MS100 would not include a temperature
sensor, as this integrated sensor has been discontinued. For this reason, we decided to not interrupt the recording of the magnetic field measurements to troubleshoot the issue with the integrated sensor but rather to add the following two items in our
setup:

- BME280-3.3, a sensor module for Arduino, which measures ambient temperature and relative humidity, by HiLetGo.
- UNO R3 BOARD, an Arduino board for the BME280 module, by Elegoo.

The BME280 sensor module was encapsulated in a 3D-printed housing, which was mounted tightly against the magnetometer sensor and connected to an Arduino UNO board. It successfully collected temperature and humidity data during the testing phase in our lab in Harvard. However, once installed on-site at the end of September, the sensor failed to provide readings. Therefore, no temperature and humidity measurements were collected on-site with this sensor. Subsequent troubleshooting in

our lab showed that the issue was caused by the increased capacitance and signal degradation resulting from the long cable length (8 meters) between the BME280 sensor and the Arduino. The addition of external pull-up resistors to the Arduino board, resolved this issue. The on-site testing of this modified version of the temperature and humidity module will take place during future deployments.

Author contributions. All authors were involved in selecting the main instrumentation elements. AD, EK, AW and FV were responsible for ordering and receiving the various components. AD manufactured all custom components and carried out the design and assembly of the experimental setup, with input from FV. LD, AW, EM, EK and FV contributed to the recording and storage of the data. AW performed the calibration of the magnetometer at the Boulder magnetic observatory, with support from FV and SL. EK was responsible for the deployment and maintenance of the station, with support from AD and FV. FV processed and interpreted the data, with contributions from AW, SL, WW and LD. FV prepared the manuscript, with contributions from all co-authors.

Competing interests. The authors declare that they have no conflict of interest.

Acknowledgements. We would like to thank Kader Telali for his help and guidance concerning the calibration and deployment of our instrumentation. We are thankful to the group of the Boulder USGS magnetic observatory for providing us access to their infrastructure and hands-on assistance for the calibration of our magnetometer. Ludo Letourneur has been a source of support at all stages of this project, from offering pre-sales advice to answering all of our questions concerning the calibration, operation and deployment of the magnetometer. We are thankful to Andriy Fedorenko for providing feedback on an earlier version of this manuscript and to Byron Wagner for his advice on the use of ferrites. We also thank two anonymous reviewers and the Associate Editor Lev Eppelbaum for useful feedback that helped improve our manuscript. FV is currently affiliated with Aurora Technology B.V. for ESA - European Space Agency, European Space Astronomy Centre (ESA/ESAC), but she conducted this work as a volunteer for the Galileo Project.

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
