# Peer review of "The deployment of a geomagnetic variometer station as auxiliary instrumentation for the study of Unidentified Aerial Phenomena"

_EGUsphere, 2025_

## Author Comment (AC2)

Revised Figure 4:

[Figure]

Figure 4: The results of an empirical evaluation of our temperature calibration, during a 1-hour recording under natural temperature variation. (a) The intensity of the raw uncalibrated data, plotted over time. (b) The results of the calibration. Black line: the intensity of the 1-sec uncalibrated vector data. Magenta line: the intensity of the calibrated vector data. Grey lines: the maximum and minimum values of the intensity of the calibrated vector data, accounting for the 1-σ least-squares uncertainties of the calibration coefficients shown in Figure 3. Orange line: the scalar data. Light blue line: the scalar data, with their baseline adjusted to that of the intensity of the uncalibrated vector data. Blue line: the scalar data, with the baseline adjusted to that of the intensity of the calibrated vector data.

Revised Figure 5:

[Figure]

Figure 5: Diagram of the deployment site, showing the location of various instruments, including the magnetometer, and the distance among them. The blue circle shows the location of the hole, inside which the magnetometer was deployed. The blue rectangular shows the location of the magnetometer's data acquisition system. The yellow triangle and cyan square show the location of the Pan-Tilt-Zoom Beacon 8.0 camera (PTZ) and the all-sky camera arrays (MD) (Domine et al., 2025), respectively, which were also deployed at the site. The light yellow rectangular shows the location of the lab space.

Revised Figure 9:

[Figure]

Figure 9 : Vector magnetic field data recorded on May 9, 2024, a magnetically quiet day (Kp<3). Left column: Our raw data. Right column: Our 1-sec magnetic field data (black solid line), after their baseline has been adjusted to the nearest USGS magnetic observatory, which is located in Boulder, Colorado (BOU), and 1-sec magnetic field data recorded at BOU (red dashed line). From top to bottom: The magnetic field components pointing toward geographic North (X), geographic East (E), and Down (Z), respectively.

Former Figure 10 with a 10 sec zoom-in inset:

[Figure]

Revised Figure 10:

[Figure]

Figure 10: Raw temperature data, recorded with the integrated temperature sensor on May 9, 2024, with a sampling rate of 1612.9 Hz. The inset shows a 300 data points zoom-in. Temperature readings vary by less than 0.3 ∘C, as a result of the thermal insulation measures we took during deployment. Similar diurnal temperature variations characterize all our data up until July 1st, which is when the temperature readings became erratic (see text for details).

Revised Figure 14:

[Figure]

Figure 14: Our magnetic field measurements showing noise in the form of spikes (see text for details). (a) Raw (i.e., unprocessed) measurements of the magnetic field component pointing North (X) on September 9, 2024. Note the spike at 13:56:39 UTC (marked by the arrow). The inset shows a 1 second long extract centered on the spike. (b) The magnetic field component pointing North (X) of 1-sec data. Black solid line shows our magnetic field data, after their baseline has been adjusted to that of BOU. The moving average filtering eliminated the spike. Red dashed line shows 1-sec magnetic field data recorded at BOU. (c-d) Same as (a-b) for the magnetic field component pointing Down (Z). Note that the moving average filtering did not eliminate the spike but only reduced its amplitude.

---

## Author Response (AR1)

**Response to Reviewer 1**

We thank the Reviewer for reading and commenting on our manuscript. We provide our replies below each comment, which is shown in italics.

1. I note that the air-conditioning system at the BOU observatory was not working during calibration. Since the authors are set up near that observatory, are there plans to revisit the BOU observatory and investigate system response over a wider range of temperatures?

The specific site discussed in this manuscript is no longer operational. As we mention in Section 6 of the manuscript, the magnetometer remained deployed only until September 26, 2024.

Given the temperature stability achieved by means of the insulation measures we took and the fact that we are using the magnetometer as a variometer and not for absolute measurements, we do not anticipate the need for calibrating our magnetometer over a wider range of temperatures. Nevertheless, we will keep monitoring the temperature variations and if need be we will seek ways to calibrate it at the closest available facility.

2. Are there plans to make the data available to the scientific community in the style of (say) Intermagnet or SuperMag? This is important. Other investigators will make use of the data, and, in the end, the project represented in this manuscript will find indirect support from the wider scientific community.

Yes, all our data are meant to be publicly shared. We have already uploaded at Zenodo all the raw magnetic field and temperature data collected over the days discussed in the manuscript, and we are happy to share the rest of our data with anyone interested. Currently, due to limited resources, we cannot make all data directly available. By default, we process the data we consider interesting based on a variety of factors. For example, in the case of this manuscript, we considered interesting the data that were representative of the performance of the magnetometer. But we are happy to allocate resources to respond to specific data requests.

**Response to Reviewer 2**

We thank the reviewer for the encouraging feedback and the detailed comments that help us improve the manuscript. We provide our responses below each comment, which is shown in italics.

 Page 5: The discussion of the magnetometer / ADC interface is a bit short, e.g. it is nice to be told the noise floor of the magnetometer, but if it is poorly interfaced to a insufficient ADC, this noise floor is never reached. Please elaborate a bit here. How far is the experimental noise floor from the theoretical noise floor? We have added information about the specs of the ADC in the revised manuscript. More precisely, we addressed this comment by doing the following:

**Section 2**

We now term NI-9239 the Data Acquisition Module (DAQ) and explain that it contains an ADC rather than being just an ADC.

We provide the following specifications for the DAQ and its ADC:

"The DAQ contains a 24-bit delta-sigma analog-to-digital converter and has an input range of +/- 10 V. The DAQ allows for sampling rates,  $f_s$ , that range from 1.613 to 50 kHz, has an alias-free bandwidth of 0.453 x  $f_s$ , and an input-referred noise of 70  $\mu$ Vrms or equivalently 700 pT. Assuming that the input-referred noise corresponds to the highest possible sampling rate, the expected noise in our raw measurements, given that we sample at the lowest possible  $f_s$ , is approximately 120 pT. This makes the noise floor of the DAQ the main contributor to the expected noise of our setup."

To allow the reader to follow the conversion from V to T, we now specify the sensitivity of the magnetometer: "The measurement range of the magnetometer is +/- 100  $\mu$ T, its sensitivity is 1 V per 10  $\mu$ T,...."

We also provide the noise floor of the PSU: "The PSU has a noise floor of  $

Figure 9: Vector magnetic field data recorded on May 9, 2024, a magnetically quiet day (Kp

Figure 4: The results of an empirical evaluation of our temperature calibration, during a 1-hour recording under natural temperature variation. (a) The intensity of the raw uncalibrated data, plotted over time. (b) The results of the calibration. Black line: the intensity of the 1-sec uncalibrated vector data. Magenta line: the intensity of the calibrated vector data. Grey lines: the maximum and minimum values of the intensity of the calibrated vector data, accounting for the 1-σ least-squares uncertainties of the calibration coefficients shown in Figure 3. Orange line: the scalar data. Light blue line: the scalar data, with their baseline adjusted to that of the intensity of the uncalibrated vector data. Blue line: the scalar data, with the baseline adjusted to that of the intensity of the calibrated vector data.

• Figure 5: What is MD? I suggest to also add the distance between the magnetometer and the camera as this is not directly clear from the picture.

We have included the information about the distance in the figure (9.4 m), and we have included the initials PTZ and MD in the caption. In particular, MD corresponds to the all-sky camera arrays.

This is the revised Figure 5:

• Figure 10 could be discussed in slightly more detail. Clearly, the instrument is very well insulated as seen by the small overall change in temperature, but the plotted data curve is very "thick". I suggest adding an inset with a zoom on a few seconds of data to make the data quality and any structure in it more visible. Please also add a comment about the "thickness" of the plot. Is it explained by the electronic properties of the sensor or are external noise sources in play?

We implemented the suggestion of the reviewer to add to Figure 10 an inset with a 10 sec zoom:

However, we consider that this inset still does not allow to discern any structure in the data. The raw data have been obtained at a sampling rate of 1612.9 Hz, and we need to go down to a resolution of a few hundred data points to start discerning any structure in the data. For this reason, we updated Figure 10 to include an inset of a 300 data points zoom-in. This is the revised Figure 10, along with the revised caption:

Figure 10: Raw temperature data, recorded with the integrated temperature sensor on May 9, 2024, with a sampling rate of 1612.9 Hz. The inset shows a 300 data points zoom-in. Temperature readings vary by less than 0.3 °C, as a result of the thermal insulation measures we took during deployment. Similar diurnal temperature variations characterize all our data up until July 1st, which is when the temperature readings became erratic (see text for details).

Moreover, we added a comment about the thickness of the plot:

"We note that the RMS noise in the raw temperature data shown in Figure 10 is approximately 0.07 °C. Although we do not have information about the noise specifications of the integrated temperature sensor, the observed noise level is consistent with typical industrial temperature sensors."

And we also clarify at the beginning of the respective paragraph that these are raw data obtained with a sampling rate of 1612.9 Hz:

"Figure 10 shows the raw temperature measurements, obtained with a sampling rate of 1612.9 Hz by the integrated temperature sensor of our magnetometer, during the day of May 9th, 2024. The inset shows a 300 data points zoom-in."

• Figure 14: I suggest adding insets with zooms on the spikes, so the shape of the spikes can be seen.

We implemented the recommendation. This is the revised Figure 14:

Figure 14: Our magnetic field measurements showing noise in the form of spikes (see text for details). (a) Raw (i.e., unprocessed) measurements of the magnetic field component pointing North (X) on September 9, 2024. Note the spike at 13:56:39 UTC (marked by the arrow). The inset shows a 1 second long extract centered on the spike. (b) The magnetic field component pointing North (X) of 1-sec data. Black solid line shows

our magnetic field data, after their baseline has been adjusted to that of BOU. The moving average filtering eliminated the spike. Red dashed line shows 1-sec magnetic field data recorded at BOU. (c-d) Same as (a-b) for the magnetic field component pointing Down (Z). Note that the moving average filtering did not eliminate the spike but only reduced its amplitude.

Page 21: The discussion of the origins of spikes is a bit short. Are there other
potential sources of spikes in magnetometer data, e.g., could cosmic rays be the
culprit? Is spikes like this this a common feature for magnetic observatories? On
page 6 a 160 m distant road is mentioned. Is there any correlation to traffic on this
road or can any other noise be attributed to traffic?

To address this comment by the reviewer, we have added this paragraph in the Discussion section:

"There are many different sources that can give rise to spikes in magnetometer data, and it is possible that not all of the spikes in our data have a common source. The spikes that have a duration of less than a second are most probably not of natural origin, but longer spikes can also be of artificial origin, like digital errors. For example, we see in Figure 9e that the spike occurs just after a change in the gain. Some spikes could be due to electromagnetic interferences, which themselves have a large variety of causes. The road at 160 m distance from our site could be the source of some interference, especially whenever large vehicles like trucks were passing by. While we are not able to interpret all spikes in our raw data, we expect that some will be eliminated after we install the ferrites recommended by NI for electromagnetic compatibility compliance when using NI-9239 (i.e., our DAQ), as opposed to the generic, cost-effective ferrites we used in this deployment."

INTERNAGNET observatories typically do not publish data at a higher resolution than 1-sec data, and these data are typically cleaned before being published.